# Mercury Methylation Potentials in Sediments of an Ancient Cypress Wetland Using Species-Specific Isotope Dilution GC-ICP-MS

**DOI:** 10.3390/molecules27154911

**Published:** 2022-08-01

**Authors:** Derek D. Bussan, Chris Douvris, James V. Cizdziel

**Affiliations:** 1Department of Chemistry, Eastern Kentucky University, Richmond, KY 40475, USA; 2Department of Chemistry and Biochemistry, University of Mississippi, University, MS 38677, USA; 3Department of Biological & Chemical Sciences, New York Institute of Technology, Old Westbury, NY 11568, USA; cdouvris@nyit.edu

**Keywords:** mercury, methylmercury, methylation and memethylation rates, wetlands, ICP-MS

## Abstract

Wetlands are of a considerable environmental value as they provide food and habitat for plants and animals. Several important chemical transformations take place in wetland media, including the conversion of inorganic mercury (Hg) to monomethylmercury (MeHg), a toxic compound with a strong tendency for bioconcentration. Considering the fact that wetlands are hotspots for Hg methylation, we investigated, for the first time, Hg methylation and demethylation rates in an old growth cypress wetland at Sky Lake in the Mississippi Delta. The Sky Lake ecosystem undergoes large-scale water level fluctuations causing alternating periods of oxic and anoxic conditions in the sediment. These oscillating redox conditions, in turn, can influence the transformation, speciation, and bioavailability of Hg. In the present study, sediment cores from the wetland and Sky Lake itself were spiked with enriched stable isotope tracers of inorganic Hg and MeHg and allowed to incubate (in-situ) before freezing, sectioning, and analysis. Methylation rates (day^−1^) ranged from 0.012 ± 0.003 to 0.054 ± 0.019, with the lowest rate in the winter and the highest in the summer. Demethylation rates were about two orders of magnitude higher, and also greater in the warmer seasons (e.g., 1.84 ± 0.78 and 4.63 ± 0.51 for wetland sediment in the winter and summer, respectively). Methylation rates were generally higher in the open water sediment compared to wetland sediment, with the latter shaded and cooler. Both methylation (*r* = 0.76, *p* = 0.034) and demethylation (0.97, *p* = 0.016) rates (day^−1^) were positively correlated with temperature, but not with most other water quality parameters. MeHg concentration in the water was correlated with pH (*r* = 0.80, *p* < 0.05), but methylation rates were only marginally correlated (*r* = 0.71). Environmental factors driving microbial production of MeHg in the system include warm temperatures, high levels of labile natural organic matter, and to a lesser extent the relatively low pH and the residence time of the water. This study also provides baseline data that can be used to quantify the impacts of modifying the natural flow of water to the system on Hg methylation and demethylation rates.

## 1. Introduction

Methylmercury (MeHg) is a neurotoxin species produced primarily by sulfate reducing bacteria in aquatic sediments that readily biomagnifies up freshwater and marine food chains [1]. Consumption of fish containing high levels of MeHg can lead to severe adverse health effects in both humans and wildlife [2]. To understand the factors controlling the environmental concentration of MeHg in natural water bodies, there is a need to measure both inorganic (Hg^+2^) and organic (MeHg) forms, as well as the relative rates of methylation and demethylation. Thus, enriched stable isotopes have been used to simultaneously study the methylation and demethylation reactions [3].

Mercury entering naturally occurring aquatic systems stems from both natural and anthropogenic sources, with the latter primarily from combustion of fossil fuels, as well as contaminated from legacy Hg mines, soda-PVC plants, artisanal gold mining, etc. [4]. Inorganic Hg can be transformed to MeHg by abiotic and biotic methylation processes [4,5,6]. The majority of Hg methylation is carried out by sulfate-reducing bacteria (SRB), with a minor contribution from Fe-reducing bacteria. These bacteria use vitamin B-12 for transferring a CH_3_ group via radical reactions [7].

Wetlands are an important resource because they provide many positive environmental benefits, including habitat for wildlife, buffering for flood-prone areas, and a natural filtering effect of some water impurities [8,9]. Wetlands also play a critical role in the transformation of Hg within landscapes and are considered hotspots for Hg methylation, in part because SRB thrive in wetland sediments [10]. Wetlands have been shown to be net sources of MeHg to ecosystems and may, in part, explain the high concentrations of Hg found in fish in remote and near-pristine regions [11].

MeHg diffuses out of the microorganism and enters the water column where it readily accumulates in plankton and biomagnifies in the food web where it can have ecological and human health impacts. Environmental factors that seem to promote methylation, all of which can be found in wetlands, include warm temperatures, low pH, the presence of humic materials, and long water residence time [12,13]. Previous studies have shown that humic materials can accumulate heavy metals, including Hg, and have been shown to increase natural bacterial assemblages, serving as a carbon and energy source [14,15].

As wetlands are documented to exhibit elevated MeHg production compared to other aquatic habitats, it is of paramount importance to assess Hg methylation in different types of wetlands and within important watersheds. Considering the significance, we set out to: (1) measure methylation/demethylation rates seasonally in an ancient cypress wetland and adjacent open water areas in Sky Lake, (2) correlate methylation/demethylation rates with ancillary data (pH, dissolved oxygen, temperature, conductivity, etc.), and (3) determine the factors driving Hg methylation in the system. Finally, this study also provides baseline data that can be used to quantify the impacts of altering the natural flow of water to the system on Hg methylation and demethylation rates. Such changes may occur with climate change or more abruptly with changes in land use.

## 2. Materials and Methods

### 2.1. Site Descriptions

Sky Lake is an oxbow lake located in the Mississippi Delta between the Yazoo and the Big Sunflower Rivers (Figure 1). A wetland associated with the lake contains one of the few old growth Cypress forests remaining in the world, 10 km from the town of Belzoni (Figure 2). The site has an elevated boardwalk that stands atop a portion of the wetland. The cypress wetland at Sky Lake undergoes large-scale water level fluctuations that are thought to affect the redox conditions in the sediment resulting in alternating periods of oxic and anoxic conditions. These oscillations in redox conditions may influence the methylation and demethylation rates, which affect speciation and bioavailability of Hg in the system. We were granted permission to access the site and setup instrumentation.

The UM Field Station is located 11 miles northeast of the UM Oxford campus off County Road 202. The Field Station is located on 740 acres of land and consists of 200 experimental ponds ranging from 0.1 to 2 acres. Ponds are typically about 1 m in depth. The UM Field Station site was chosen as a control site due to the water level stability and was conducted at pond 179 (Figure 3). A prior study with sediments from this site observed the effect of activated carbon, biochar, and humus/manure on Hg methylation rates [14]. All maps were provided by Google Map image except Figure 3 which comes from the Mississippi Department of Transportation.

### 2.2. Enriched Stable Isotopes of Hg and Reference Materials

Enriched isotopes of ^199^HgO, ^200^HgO, and ^201^HgO were purchased from Oak Ridge National Laboratories (ORNL) (Oak Ridge, Tennessee, TN, USA) and dissolved in 10% optima grade nitric acid. MeHg containing enriched isotopes was subsequently synthesized using the methylcobalamine method [16]. All other reagents used were reagent grade. Ultrapure water (>18 ΜΩ cm^−1^) was obtained from a Barnstead Nanopure Diamond system. For reference materials, we used DORM-3 (dogfish muscle) for total Hg, and CC-580 (Estuarine Sediment) for MeHg. DORM-3 was obtained from the National Research Council of Canada (Ottawa), and CC-580 was obtained from the European Commission Institute for Reference Materials and Measurements (IRMM) in Belgium.

### 2.3. Sediment Collection and Processing

Two sediment cores were collected seasonally from both wetland (W) and open water (OW) areas at Sky Lake during the years 2013–2014. MeHg production is related to temperature and microbial activity, and has been shown to vary seasonally [15,17]. Wetland sediments are typically anoxic or become anoxic within the first cm or less, and thus Hg methylation is generally highest in the top layers of the sediment [15]. We chose to measure Hg methylation rates in the top 6 cm of the sediment, but on occasion even deeper. Sediment cores were collected from Sky Lake and the Biological Field Station using plastic core liners (PVC 2″ × 36″) purchased from AMS, Inc. Holes were drilled every 1 cm down the length of each tube to allow enriched isotopic tracers (Me^199^Hg^+^ and ^200^Hg^2+^) to be injected throughout the sediment core for the tracer study (Figure 4). The holes were filled with clear waterproof silicone.

Two (duplicate) sediment cores were taken at each sample site. Stable isotope spike solutions of ^200^Hg^2+^ and ^199^CH_3_Hg^+^ were prepared at concentrations ranging from ~10–100% of ambient concentrations by diluting the standard with bottom water from the site. The mixture was left to equilibrate for an hour prior to usage to attain natural conditions/speciation [18]. Approximately 0.2 mL of spike solution was injected into each 1 cm increment down to approximately 20 cm (Figure 4). After spiking, the sediment cores were placed back into the wetland and open water sites to incubate for 2 h before being transported in a cooler to the laboratory. At the lab, a modified wood extruder was used to extrude the cores onto a plastic cutting board. A stainless-steel knife was used to section the top 6 cm of the core and to divide the top 6 cm into 0–2, 2–4, and 4–6 cm sections. The samples were taken wearing gloves and stored overnight in a −80 °C freezer and subsequently lyophilized for 5 days.

### 2.4. Extracting and Isolating MeHg from the Sediment

Before distillation, all equipment used was acid-washed in a 15% HCl bath, rinsed with Ultrapure water, and dried under a laminar flow hood. The acetate buffer and 1% sodium tetraethyl-borate were prepared according to EPA Method 1630 [19]. The sediment extraction and distillation procedure is based on Method 1630 and can be found in the Appendix A.

### 2.5. Determination of MeHg Species in Sediments by GC-ICP-MS

To determine the amount of MeHg in the sediments, GC-ICP-MS was utilized as described by Lambertsson [20]. Briefly, a GC column from a purge and trap Tekran 2700 MeHg analyzer was used for the separation of the different species of Hg. The effluent coming from the atomic fluorescence cell of the Tekran 2700 is coupled via a Teflon line to the sample injector to the ICP-MS. The isotopes of Hg were detected by the electron multiplier. Prior to analysis, the solution was ethylated such that all Hg compounds became volatilized although Hg^0^ is volatile by itself. The mercuric ion becomes diethyl Hg, and MeHg becomes MeEtHg. The Hg species were separated out by the GC that is contained in the Tekran 2700. The Hg species flowed through the atomic fluorescence detector, and were introduced into the plasma of the ICP-MS which is a technique, that provides excellent separation. A typical GC-ICP-MS chromatogram is shown in Figure 5. Each data point on the chromatogram is a mass spectrum that was generated from the ICP-MS. The chromatogram shows that when Me^201^Hg is spiked into a sample mixture, the resulting ratios of 201/202 Hg are changed to 3.40. The naturally occurring ratio of 201/202 is 0.44 [21].

### 2.6. Determination of Total-Hg and Hg Isotopes in Sediments by DMA-ICP-MS

DMA coupled to ICP-MS offers several advantages over the DMA alone for the determination of mercury in methods that require distinguishing the individual isotopes [22]. These include increased sensitivity, lower detection limits, decreased potential for sample contamination, and applicability to Hg stable isotope tracer studies. As mercury isotopes are required for the methylation and demethylation calculations, total Hg concentrations and Hg isotopes in sediment were determined by DMA-ICP-MS as described in Bussan et al. [22]. Briefly, sample boats were cleaned prior to use by running the boats as blanks in the DMA. Concentrations were calculated using the single spike isotope dilution equation using ^201^Hg^2+^ and ^201^CH_3_Hg^+1^ (Equation (1)) [23]. Ambient MeHg concentrations in the sediment were also determined by GC-ICP-MS using the same equation
(1)$$C_{N}=C_{spike}\;\frac{M_{s}W_{N}\;}{M_{N}W_{S}}\left[\frac{Ab_{S}^{A}-R_{m}Ab_{S}^{B}}{R_{m}Ab_{N}^{B}-Ab_{N}^{A}}\right]$$

The abbreviations of the constants are as follows: *C_n_*: concentration of sample; *C_spike_*: concentration of spike; *M_s_*: weight of the spike; *M_n_*: weight of the sample; *W_N_*: atomic weight of the element in the sample; *W_S_*: atomic weight of the element in the spike *Ab^A^_s_*: abundance of *A* in the spike; *Ab^B^_s_*: abundance of *B* in the spike; *Ab^A^_N_*: abundance of *A* in the sample; *Ab^B^_N_*: abundance of B in the sample; and *R_m_*: ratio of the mixture.

### 2.7. Determination of Hg Methylation and MeHg Demethylation Rates

Hg methylation and demethylation rates were determined following Hintelmann et al. [18,23]. Briefly, to determine the amount of methylated and demethylated Hg at least three isotopes of Hg need to be monitored. One isotope represents the newly produced MeHg from the added inorganic Hg tracer (^200^Hg in this study); another represents the demethylation of the added MeHg tracer (Me^199^Hg); and the third represents changes in the MeHg concentrations derived from the Hg originally present in the sample (e.g., ^202^Hg) [1].

The net production of methylation is represented by Equation (2).
(2)d[CH3H Xg+]dt=km[H Xg2+]−kd[CH3H Xg+ ]
where [*CH*_3_^X^*Hg*^+^] = concentration of *CH*_3_^X^*Hg*^+^ newly generated from the *^X^Hg*^2+^ tracer (in ng/g), *k_m_* = specific methylation rate constant (in d^−1^), [*^X^Hg*^2+^] = concentration of added *^X^Hg*^2+^ (in ng/g), *k_d_* = specific demethylation rate constant (in d^−1^), and *t* = incubation time (in days). By having a large excess of *XHg*^2+^, we can solve for the specific methylation constant assuming a pseudo-first-order reaction; Equation (2) becomes Equation (3):(3)km=[CH3H 199g+][H 199g2+]t

Likewise, when *CH*_3_^199^*Hg*^+^ is spiked to the sediment, the ^200^*Hg*^2+^ from the demethylation is negligible, and Equation (3) reduces down to:(4)d[CH3H 199g+]dt=−kd[CH3H 199g+] 

Integration of Equation (4) yields Equation (5), which can be used to determine the demethylation rate constant (*k_d_*).
(5)[CH3H 199g+]=[CH3H 199g+] 0 e−kdt

### 2.8. Instrumental Parameters

The DMA-ICP-MS instrumental parameters used in this study are given in Table 1.

### 2.9. Water Quality Parameters

Water quality parameters were measured with a portable YSI Professional Plus meter. Dissolved oxygen (DO) (mg L^−1^), pH, total dissolved solids (TDS) (mg L^−1^) using the default instrument conversion constant, oxidation reduction potential (ORP) (mV), and temperature (°C) were measured (Table 2). Total suspended solids (TSS) (mg L^−1^) were measured using a portable suspended solids analyzer (InsitelG, model 3150). Sulfide (S^2-^) (mV) was measured with an Orion silver/sulfide ion selective electrode using Pb (ClO_4_)_2_ as the titrant. Loss on ignition (LOI) was calculated using a relationship established in our previous work between LOI and total carbon and which is specific for the sampling sites [24].

### 2.10. Statistical Analysis

Statistical evaluation was performed using STAT PLUS 2009 software. Data were evaluated for correlations using the Pearson Linear correlation test. In addition, *p*-values or observed significance values were calculated for the different parameters measured. A *p*-value < 0.05 is considered to be of statistical significance. Univariate analysis of variance (ANOVA) with post hoc Tukey HSD test was also used as a statistical test to assess the differences among groups based on a set of dependent variables.

## 3. Results and Discussion

A summary of Hg methylation and demethylation rates is given in Table 3, while total Hg, MeHg, and organic matter (LOI) for the sediment and MeHg data for the water is provided in Table 4. Ancillary water quality data, including temperature, dissolved oxygen, total dissolved solids, oxidation reduction potential, pH, total suspended solids, conductivity, and sulfide concentration is given in Table 2. These results are discussed in the subsections below.

Sediment total Hg on a dry weight basis ranged from 12.5 ng g^−1^ to 189.0 ng g^−1^ at the open water site and from 20.0 ng g^−1^ to 121.6 ng g^−1^ at the wetland water site. The large variation of total mercury may be attributed to the rate of conversion of inorganic mercury being converted to methyl mercury by microscopic organisms in sediments and water. Sediment MeHg ranged from 0.36 ng g^−1^ to 1.34 ng g^−1^ at the open water site and ranged from 0.43 ng g^−1^ to 1.43 ng g^−1^ at the wetland water site. These results are similar to Hugget et al. who reported sediment Hg levels ranging from 27 ng g^−1^ to 168 ng g^−1^ in three Mississippi reservoirs (Enid, Sardis and Grenada Lakes) [25]. There are no major source points in either of these two different watersheds, so the likely source of the Hg is atmospheric deposition from both natural and distant anthropogenic sources, along with some leaching of Hg from the mineral environment.

Hall et al. (2008) sampled surface water and pore water in a variety of wetlands in southern Louisiana and the Gulf of Mexico region between August of 2003 and May 2005. Their study revealed that total Hg and MeHg concentrations in surface waters were appreciably greater in freshwater and brackish wetlands, as these sites are expected to support microbial methylation of Hg, while the values were lower in marine wetlands and brackish and marine open water systems. In particular, for surface waters, they reported MeHg levels as high as 2.15 ng L^−1^ and total Hg levels as high as 5.58 ng L^−1^ [26]. Both values are higher than the ones reported in the present study. In addition, the authors determined that MeHg concentrations were greatest in wetland water compared to non-vegetated regions of the lake and river mainstems. However, we found slightly lower concentrations for MeHg in water at the wetland site (5.93 ± 0.87 pg g^−1^) compared to the open water site (7.41 ± 0.83 pg g^−1^), although we have too few data to make a robust statistical comparison.

Hg methylation rates were lowest in the winter and highest in the summer for the open water site (Figure 6). This is likely related to higher microbial activity with higher temperatures [27]. Indeed, we found Hg methylation rates and temperature were correlated (*r* = 0.76, *p* = 0.034) despite the relatively low K_m_ in the summer for the wetland (Figure 7 and Figure 8). As reported by others, temperature plays a major role for seasonal differences in Hg methylation rates [28,29]. In our study, the difference between methylation rates for the open water and wetland sites were likely exacerbated by the dense foliage and tree canopy of the forested wetland. The water temperature of the shaded wetland was substantially cooler than the unshaded open water site. In the winter, when foliage coverage is not a significant factor in the wetland, the methylation rates were found to be virtually identical. The same was found to be true in the early spring when wetland foliage had not yet peaked.

Methylation rates were also generally higher in the open water sites compared to the wetland sites, except in winter and spring where they were nearly identical.

The demethylation rates increased as temperature increased (Figure 9). Demethylation can occur through both oxidative and reductive processes. The end products are what define the type of process that happens, either CO_2_ (oxidative demethylation) or CH_4_ (reductive demethylation). Marvin-Dipasquale et al. found that reductive demethylation occurs in highly contaminated sediments as a detoxification process and is performed by bacteria containing the mer-operon [30,31,32,33]. The microbes in the biotic reaction can cleave the methyl group from MeHg resulting in CH_4_ and Hg^2+^ products using the enzyme organomercurial lyase [34].

A relationship between Hg methylation rates and ancillary data using a Pearson’s correlation matrix on the wetland and open water sites revealed the relationship between K_m_, Hg, MeHg, and ancillary data (Table 5). As noted, temperature was highest during the summer months (29.3 °C, open water and 22.4 °C, wetland water) and correlated positively with K_m_ rate values (*r* = 0.76, *p* = 0.034). There was also a positive correlation between pH vs. MeHg (*r* = 0.80, *p* = 0.03), and a negative correlation between pH vs. K_m_ rate values (*r* = −0.36, *p* = 0.65). Xun et al. similarly observed that a decrease in pH would result in specific rates of Hg methylation increasing [35]. The authors hypothesized that an increase in methylation at a lower pH could be due to higher uptake of Hg^+2^ by the methylating bacteria. The process of Hg uptake differs between species of microorganism, meaning that one species incorporates Hg primarily by passive diffusion, while another species by facilitated diffusion [35,36].

### Comparison of Hg Methylation Rates with Literature Values

The values for K_m_ (day^−1^) for this study generally overlap with the range found in previous studies at other locations [1,36,37]. They also tend to be on the higher side, with mean rates higher than those found in northern Ontario, in the northwest Atlantic Ocean, and in the east Atlantic Ocean (Figure 10). As explained previously, temperature plays an important role in the methylation process. The samples from this study were obtained in the Southeast US, where the climate is warmer than the other study sites.

## 4. Conclusions

The processes that drive Hg methylation and MeHg demethylation in wetland sediments are of paramount importance as wetlands, due to their unique biochemical conditions, are known to have increased MeHg production compared to other aquatic environments. The study of these conditions and their influence on net MeHg production is important as MeHg can have profound effects on ecosystems. Here, we measured, for the first time, Hg methylation potentials in a unique and ancient cypress wetland with the aim of better understanding the complex factors affecting in situ MeHg production. As a result, a number of important data was collected, analyzed, and presented in this report. These include the amounts of MeHg in wetland sediments measured by GC-ICP-MS, which ranged from 0.36 to 1.43 ng g^−1^, and the amounts of total Hg in sediments by DMA-ICP-MS, which ranged from 12.5 to 189.0 ng g^−1^. In addition, Hg methylation and demethylation rates were assessed and found to vary seasonally with the highest rates in the summer (0.034 day^−1^) and the lowest in the winter (0.013 day^−1^). To help determine the factors driving Hg methylation in the system, correlations between methylation and demethylation rates with ancillary data were evaluated. Most critical was temperature, as the temperature increased the Hg methylation rate increased. For the open waters of Sky Lake, Hg methylation rates ranged from 0.012 to 0.054 day^−1^, slightly higher than the adjacent (shaded) wetland areas, which suggests that shallow waters in lakes should not be overlooked as potential sources of MeHg. Finally, as Hg methylation data are scarce in the literature, this work can serve as a baseline for future studies at this site, as well as to compare with other sites the Southeastern USA.

## Figures and Tables

**Figure 1 molecules-27-04911-f001:**
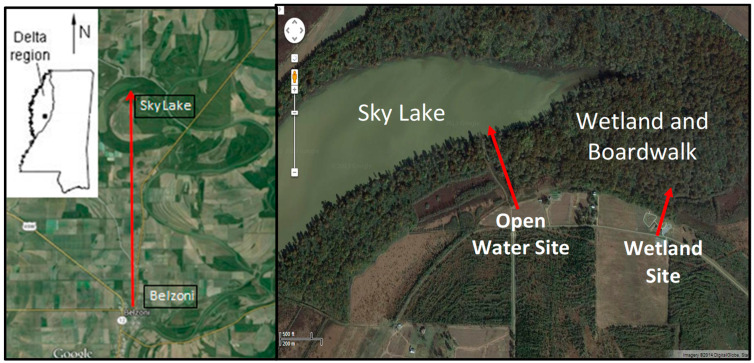
Sky Lake study area located in the Mississippi Delta. The satellite images depict the location of the open water and wetland study sites.

**Figure 2 molecules-27-04911-f002:**
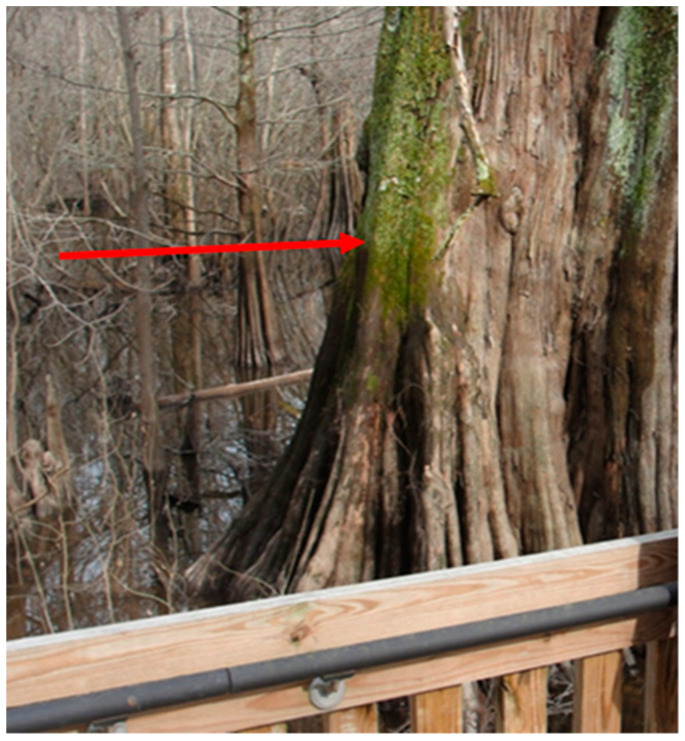
Sky Lake Wildlife Management Area. The large cypress tree in the foreground is several hundred years old. The arrow indicates a recent water level, demonstrating seasonal water level fluctuations of several meters in the wetland.

**Figure 3 molecules-27-04911-f003:**
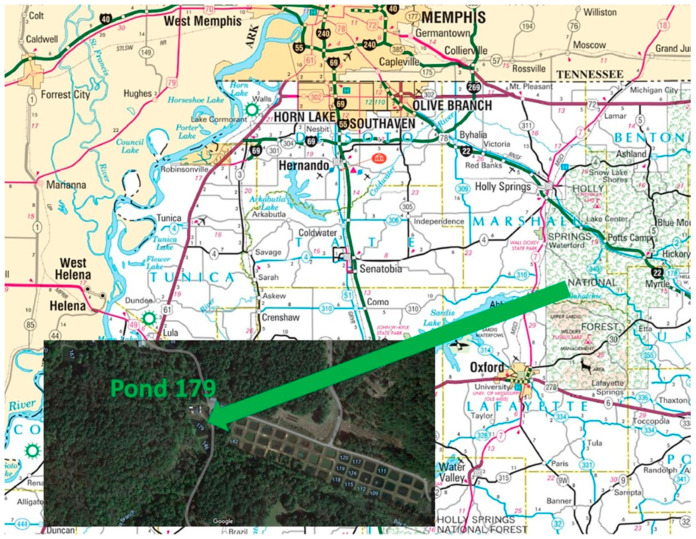
University of Mississippi Field Station—Pond 179.

**Figure 4 molecules-27-04911-f004:**
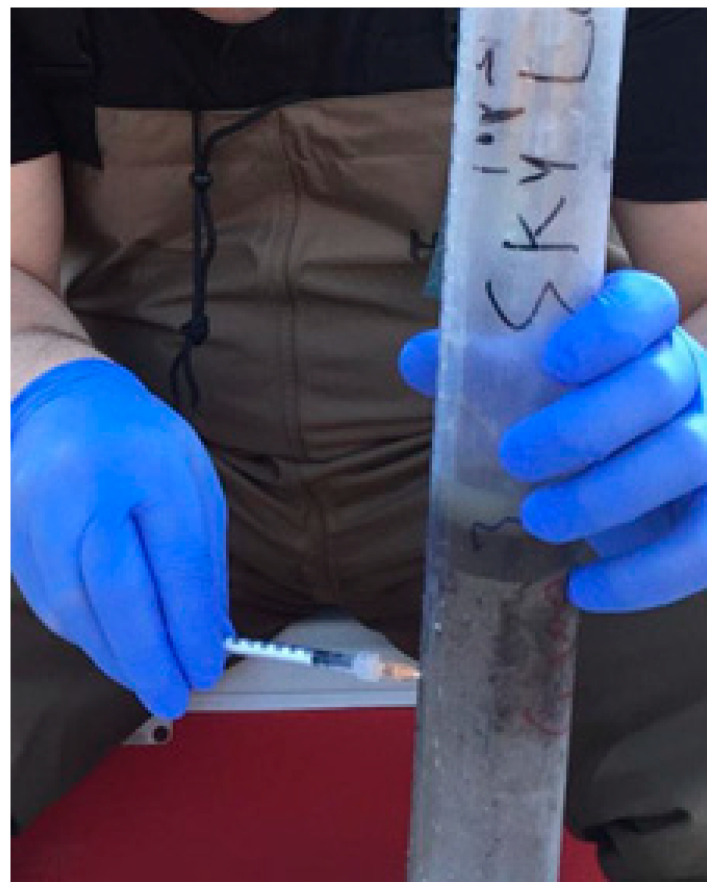
Injecting Hg tracer isotopes into a sediment core.

**Figure 5 molecules-27-04911-f005:**
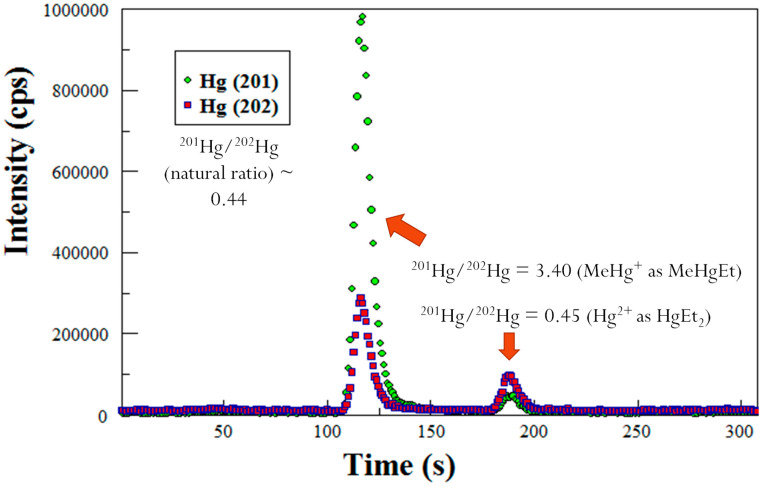
GC-ICP-MS chromatogram showing isotope ratios for Hg species: Hg^+2^ with natural abundance (peak ~ 185 s) and an enriched isotope ^201^MeHg spike (peak ~ 115 s).

**Figure 6 molecules-27-04911-f006:**
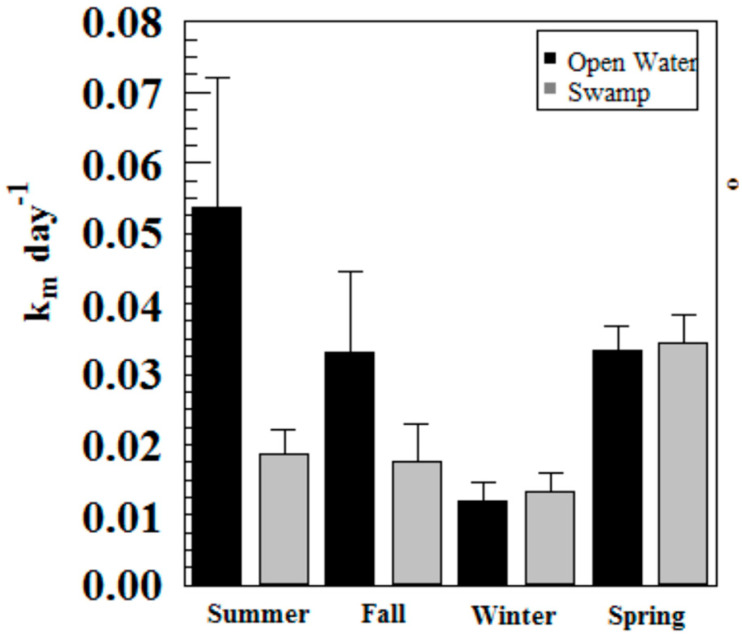
Hg methylation rates in Sky Lake by season (*n* = 3 per sample).

**Figure 7 molecules-27-04911-f007:**
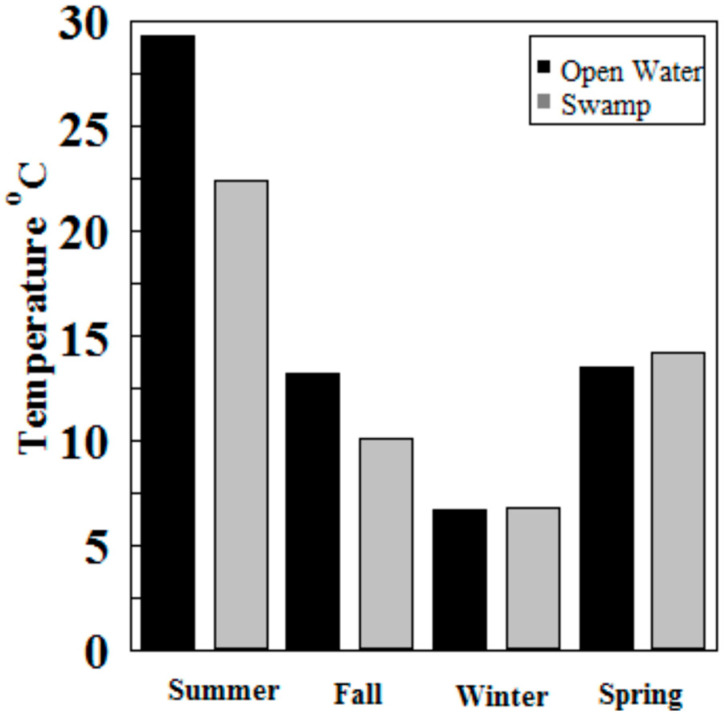
Water temperatures at Sky Lake open water and wetland sites during sampling (*n* = 3 per sample).

**Figure 8 molecules-27-04911-f008:**
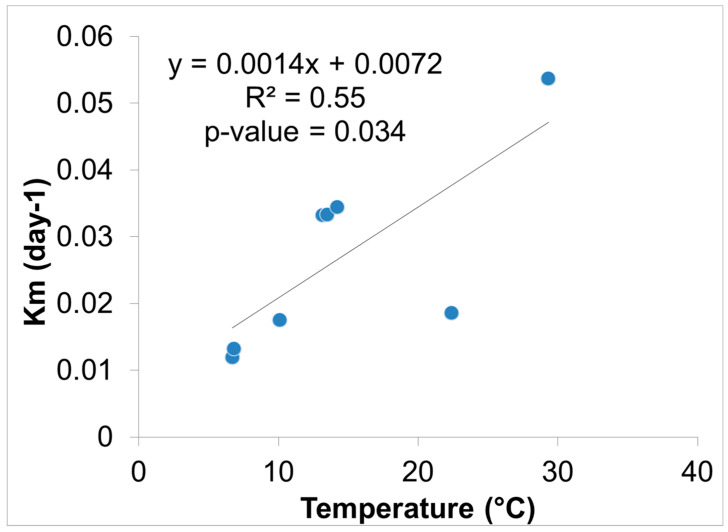
Hg methylation rate vs. ambient water temperature near the sediment surface (*n* = 8).

**Figure 9 molecules-27-04911-f009:**
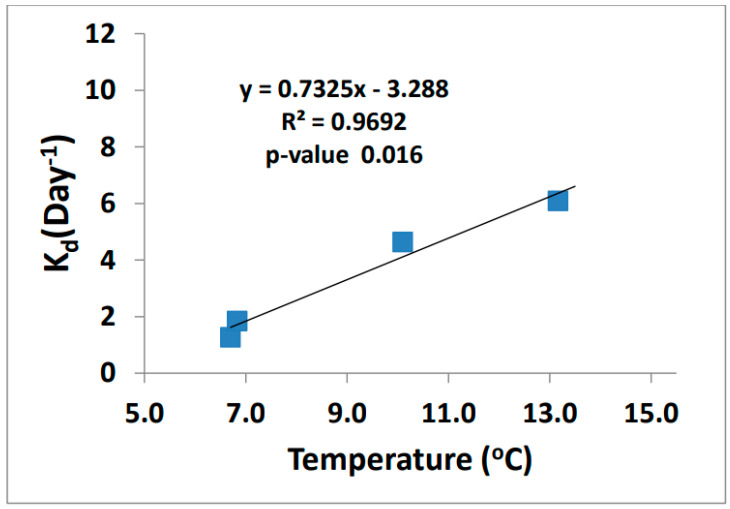
MeHg demethylation rate vs. ambient temperature (*n* = 4).

**Figure 10 molecules-27-04911-f010:**
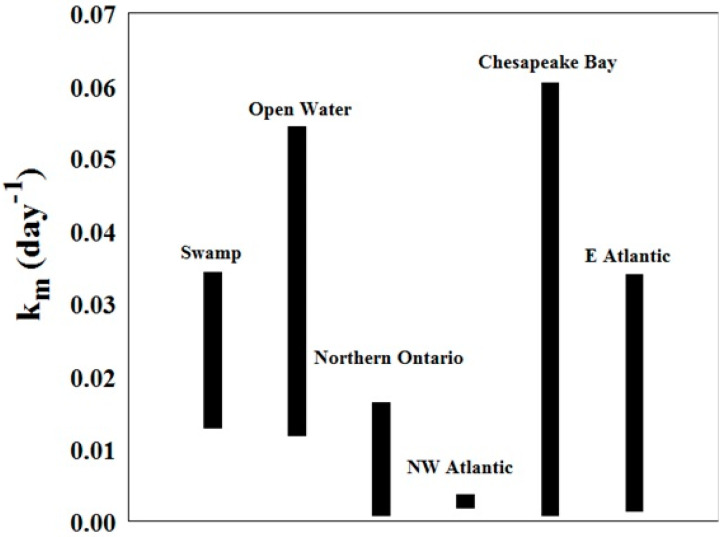
Methylation rates from this study (bars labelled swamp and open water) compared to literature values at sites along the Atlantic Coast and in northern Ontario [1,36,37].

**Table 1 molecules-27-04911-t001:** DMA-ICP-MS instrumental parameters.

**DMA Parameters**	
Gas flow	200 mL min^−1^
Drying	200 °C for 60 s
Decomposition	650 °C for 180 s
Purge 1	60 s
Amalgamator heat	850 °C for 12 s
Purge 2	45 s
**Tekran 2700 Parameters**	
Gas flow	45 mL min^−1^
Pyrolyzer	750 °C
GC Oven	75–200 °C
**Plasma Parameters**	
Cool gas flow	16 L min^−1^
Aux. gas flow	1.03 L min^−1^
Sample gas flow	1.05 L min^−1^
RF power	1280 W
**ICP-MS Data Acquisition Parameters**	
Isotopes	^201^Hg, ^202^Hg
Resolution	Low
Mass Window	5%
Points Per Peak	100

**Table 2 molecules-27-04911-t002:** Water quality parameters for Sky Lake open water and wetland areas during sampling.

Season	Location	Temp (°C)	DO (mg/L)	TDS (mg/L)	ORP (mV)	pH	TSS (mg/L)	Cond (µs/cm)	Sulfide (*µ*M)
Summer	OW	29.3	10.6	147.8	65.9	5.9	89.3	246.6	
Fall		13.2	10.5	181.7	120	7.2	169.0	216.7	3.2
Winter		6.7	5.9	157.5	−265.7	6.6	125.0	156.8	
Spring		13.5	6.6	113.9	−282.4	5.4	118.7	136.9	
Summer	W	22.4	0.7	92.1	66.4	5.3	50.3	134.6	
Fall		10.1	2.9	221.7	134.9	6.8	81.7	245.3	21.4
Winter		6.8	7.3	101.2	−258.8	6.6	27.0	100.7	
Spring		14.2	6.6	101.6	−295.5		43.0	123.8	
Spring	UMFS	18.5	11	51.8	−346.5	5.8	23.0	69.4	

OW: open water; W: wetland; UMFS; University of Mississippi Field Station.

**Table 3 molecules-27-04911-t003:** Sediment Hg methylation (K_m_) and MeHg demethylation (K_d_) rates for sediment in Sky Lake (open water and wetland areas) and the University of Mississippi’s Biological Field Station (*n* = 3; ± 1 SD) *n*, bulk samples of the same sediment.

Season	Location	K_m_ (Day^−1^)	K_d_ (Day^−1^)	Net K_m_/K_d_
Summer	OW	0.054 ± 0.019	-	-
Fall		0.033 ± 0.011	6.08 ± 4.70	0.005
Winter		0.012 ± 0.003	1.26 ± 1.09	0.009
Spring		0.033 ± 0.003	-	-
Summer	W	0.019 ± 0.004	-	-
Fall		0.018 ± 0.005	4.63 ± 0.507	0.004
Winter		0.013 ± 0.003	1.84 ± 0.776	0.007
Spring		0.034 ± 0.004	-	-
Spring	UMFS	0.12 ± 0.072		

OW = Open Water; W = Wetland; UMFS = UM Field Station.

**Table 4 molecules-27-04911-t004:** Concentrations of bulk samples of the same sediment for total Hg and MeHg in sediment and water from Sky Lake wetland and open water areas, as well as loss on ignition (LOI), an estimate of organic matter in the sediment.

Season	Location	Sediment			Water
		Total-Hg (ng g^−1^)	MeHg (ng g^−1^)	LOI (%)	MeHg (pg g^−^^1^)
Summer	OW	189 ± 22.7	0.71 ± 0.40	Average for seasons 14.7 ± 1.8	-
Fall		113.8 ± 6.1	0.98 ± 0.13		-
Winter		67.8 ± 37.6	1.34 ± 0.39		7.41 ± 0.83
Spring		12.5 ± 0.8	0.36 ± 0.08		-
Summer	W	121.6 ± 13.2	0.43 ± 0.16	Average for seasons 14.5 ± 4.5	-
Fall		116.9 ± 18.1	1.05 ± 0.49		-
Winter		94.8 ± 12.8	1.43 ± 0.49		5.93 ± 0.87
Spring		20.0 ± 1.0	0.44 ± 0.06		-
Spring	UMFS	3.7 ± 1.7	0.33 ± 0.02	-	-

OW = open water; W = wetland; UMFS = University of Mississippi Field Station.

**Table 5 molecules-27-04911-t005:** Pearson’s correlation matrix for water quality parameters measured in seasonal trends for Sky Lake, Mississippi, open water, and wetland sites. Correlations with *p*-values < 0.05 are italicized.

	**K_m_ (Day^−1^)**	**Temp (°C)**	**DO (mg/L)**	**TDS (mg/L)**	**ORP (mV)**
K_m_ (day^−1^)	1.00				
Temp (°C)	*0.76*	1.00			
DO (mg/L)	0.65	0.13	1.00		
TDS (mg/L)	0.06	−0.21	0.14	1.00	
ORP (mV)	0.34	0.52	−0.06	0.51	1.00
TSS (mg/L)	0.33	−0.09	0.48	0.50	0.11
Cond (µs/cm)	0.55	0.38	0.30	*0.82*	0.75
THg (ng/g)	0.44	0.65	0.21	0.23	0.73
MeHg (ng/g)	0.51	−0.67	0.20	0.32	−0.23
pH	−0.23	−0.56	0.38	0.71	0.19
	**TSS (mg/L)**	**Cond (µs/cm)**	**THg (ng/g)**	**MeHg (ng/g)**	**pH**
TSS (mg/L)	1.00				
Cond (µs/cm)	0.42	1.00			
THg (ng/g)	−0.22	0.61	1.00		
MeHg (ng/g)	−0.08	−0.05	0.05	1.00	
pH	0.35	0.37	0.14	*0.80*	1.00

## Data Availability

Data is available upon request.

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
