# Peer review of "Mercury Methylation Potentials in Sediments of an Ancient Cypress Wetland Using Species-Specific Isotope Dilution GC-ICP-MS"

_molecules, 2022, doi:10.3390/molecules27154911_

Round 1
Reviewer 1 Report
The study describes methylation/demethylation processes occurring in a wetland in Mississippi making use of isotopic tracers. Overall a good quality paper that deserves publication pending a moderate amount of review. I have few general comments:
1. The authors should emphasize in the Introduction what is the wide-reaching importance of the study beyond the (obvious) relevance at the local scale.
2. Some details of the experimental methods are missing. I also have doubts on some statistical treatment (definitely too few data). See specific comments in the attached pdf.
3. The source(s) of mercury remain undetermined. There is a hint to atmospheric deposition (from where?), but no real evidence
4. There is a large variation of total Hg contents (almost an order of magnitude, much more than variations in MeHg) - explanation?
5. Conclusions are not well written (see details in the pdf)
Other minor comments are reported in the attached pdf. Being the authors native speakers, I am not qualified to judge their English, but I nonetheless made suggestions on some words/sentences that sound strange to me (a few seem plain typos)

Author Response
Reviewer 1
We would like to very much thank the reviewer, we implemented almost all of his/her suggestions. We believe the manuscript significantly improved from his/her contributions. Below we specifically address the comments one by one.
- The authors should emphasize in the Introduction what is the wide-reaching importance of the study beyond the (obvious) relevance at the local scale.
As the reviewer requested, the importance was added in the beginning of the fifth paragraph (starting from line 66).
- Some details of the experimental methods are missing. I also have doubts on some statistical treatment (definitely too few data). See specific comments in the attached pdf.
We specifically one by one answered all the comments in the attached pdf. The reason the data are few is that we didn’t just focus on making measurements on a particular place over a period of time, but we intended to make measurements in the different seasons for investigating the seasonal influence on mercury methylation.
We respectfully note that determining mercury methylation/demethylation potentials (transformations) is demanding in both the field and laboratory, with each measurement taking several days of sample preparation, in addition to a distillation process that took several hours. Further, these processes involve highly toxic chemicals that need to be executed with the utmost care and thus are different from other less demanding measurements in terms of time, effort, and skill. For these reasons we had to limit our measurement campaign. Finally, this work was unfunded so the authors had limited financial resources to carry out the study. Nevertheless, we believe that the conclusions are meaningful as we focused on different seasons, which are all represented by our measurements. Despite the challenges we feel that the work makes a very important contribution to the field.
- The source(s) of mercury remain undetermined. There is a hint to atmospheric deposition (from where?), but no real evidence.
As there are no known major Hg point sources in the area, the major source is mostly likely atmospheric deposition from both natural and anthropogenic sources as well as some leaching of Hg from the local minerals. We have added some additional wording to text about this. Also, much of the anthropogenic source of mercury likely stems from the period before the establishment of environmental protection laws.
- There is a large variation of total Hg contents (almost an order of magnitude, much more than variations in MeHg) - explanation?
The most likely explanation is due to the rate of conversion of inorganic mercury to methyl mercury by microscopic organisms in sediment and water, we added this to the text (see lines 238-240).
- Conclusions are not well written (see details in the pdf) We revised all the conclusions according to the reviewer’s suggestions.
Other minor comments are reported in the attached pdf. Being the authors native speakers, I am not qualified to judge their English, but I nonetheless made suggestions on some words/sentences that sound strange to me (a few seem plain typos)
- Line 18 changed according to the reviewer’s comment.
- Line 43 changed according to the reviewer’s comment.
- Line 46 changed according to the reviewer’s comment.
- Line 78 changed according to the reviewer’s comment.
- Line 82 changed according to the reviewer’s comment.
- Line 103 – we purchased three bottles of HgO from Oak Ridge, each contained a different enriched isotope of mercury.
- Line 114 changed according to the reviewer’s comment.
- Line 132 we answered the reviewers comment (we wore gloves for handling the aliquots).
- Line 141 changed according to the reviewer’s comment.
- Line 155 changed according to the reviewer’s comment.
- Line 157 changed according to the reviewer’s comment.
- Line 181 changed according to the reviewer’s comment.
- Line 200 changed according to the reviewer’s comment.
- Line 208-209 changed according to the reviewer’s comment.
- Line 214 changed according to the reviewer’s comment.
- Line 218 changed according to the reviewer’s comment.
- Line 222 changed according to the reviewer’s comment.
- Line 232 changed according to the reviewer’s comment.
- Line 233 changed according to the reviewer’s comment.
- Line 244 changed according to the reviewer’s comment.
- Line 274 changed according to the reviewer’s comment.
- Line 279 the reason is that we focused on a measurement per season.
- Line 303 changed according to the reviewer’s comment.
- Figure 10, these superscripts were a mistake and we corrected it.
- Line 310 changed according to the reviewer’s comment.
- Line 315. We rewrote several parts and reorganized the conclusion section as the reviewer suggested
- Line 317. It is now obvious in the text why this contribution on Hg Methylation in wetlands is so important.
- Line 320. In addition to what was done, we added values found as the reviewer suggested.
- Line 327. Corrected according to the reviewer’s suggestion.
- Line 328. Corrected according to the reviewer’s suggestion.
Reviewer 2 Report
The authors have presented a nicely conducted study into the geochemistry of MeHg formation in various natural environments. The conclusions are succinct and clearly of wider interest to the research community. Fundamentally the study design is good with well-selected environments and a good structure. The writing is also to a good standard with only some minor stylistic quibbles here.
However, I have requested major revisions because I am concerned about the sample size. In the Results and discussion figures 8 and 9 both indicate positive correlations but the number of points is very small (8 and 4). The authors need to justify how such conclusions can be drawn from such small sample sizes. All statistics and figures need to have sample sizes reported next to them. If this can be addressed I am happy for them this work to be published.
Beyond this are some minor issues:
>Section 2.5 Stylistically in the methods use phrases such as "next" and "then" quite a lot these aren't required as the order is apparent from the paragraph structure.
There are multiple uses of google maps/Google Earth. These could breach copyright.
Figure 1 - what is the source of this image?
Figure 3 is not of a high enough standard for publication and should be re-produced fully by the authors not using a screenshot of google maps.
Is Figure 5 methods or results?
Finally, your references appear are always coming after the full stop not before. I think the convention is the other way round. Check this adheres to the journal's style guide?
Author Response
Reviewer 2
The authors have presented a nicely conducted study into the geochemistry of MeHg formation in various natural environments. The conclusions are succinct and clearly of wider interest to the research community. Fundamentally the study design is good with well-selected environments and a good structure. The writing is also to a good standard with only some minor stylistic quibbles here.
We very much thank the reviewer for his/her positive comments on the selection design, execution and writing of the project. We went over the text one more time and tried to make corrections on the minor stylistic errors.
However, I have requested major revisions because I am concerned about the sample size.
The reason the relatively few data is that we didn’t just focus on making measurements on a particular place over a period of time, but we intended to make measurements in the different seasons for investigating the seasonal influence on mercury methylation.
In the Results and discussion figures 8 and 9 both indicate positive correlations but the number of points is very small (8 and 4). The authors need to justify how such conclusions can be drawn from such small sample sizes.
We respectfully note that determining mercury methylation/demethylation potentials (transformations) is demanding in both the field and laboratory, with each measurement taking several days of sample preparation, in addition to a distillation process that took several hours. Further, these processes involve highly toxic chemicals that need to be executed with the utmost care and thus are different from other less demanding measurements in terms of time, effort, and skill. For these reasons we had to limit our measurement campaign. Finally, this work was unfunded so the authors had limited financial resources to carry out the study. Nevertheless, we believe that the conclusions are meaningful as we focused on different seasons, which are all represented by our measurements. Despite the challenges we feel that the work makes a very important contribution to the field.
All statistics and figures need to have sample sizes reported next to them. If this can be addressed I am happy for them this work to be published.
We report the samples sizes next to the graphs and figures as the reviewer suggested.
Beyond this are some minor issues:
>Section 2.5 Stylistically in the methods use phrases such as "next" and "then" quite a lot these aren't required as the order is apparent from the paragraph structure.
We removed these phrases as the reviewer suggested.
There are multiple uses of google maps/Google Earth. These could breach copyright.
We added a sentence (Line 99) as the reviewer suggested giving credit to Google Maps.
Figure 1 - what is the source of this image?
All the earth images are from Google Maps this indicated from text line 99, except figure 3 which comes from the Mississippi Department of Transportation.
Figure 3 is not of a high enough standard for publication and should be re-produced fully by the authors not using a screenshot of google maps.
Figure 3 was re-produced using a map from the Mississippi Department of Transportation.
Is Figure 5 methods or results?
Figure 5 is a method graph.
Finally, your references appear are always coming after the full stop not before. I think the convention is the other way round. Check this adheres to the journal's style guide?
Thank you but we read that on MDPI that the references could be of any style, here is the link:
https://www.mdpi.com/journal/molecules/instructions#references